# Plant Exosome-like Nanoparticles as Biological Shuttles for Transdermal Drug Delivery

**DOI:** 10.3390/bioengineering10010104

**Published:** 2023-01-12

**Authors:** Ye Wang, Yongsheng Wei, Hui Liao, Hongwei Fu, Xiaobin Yang, Qi Xiang, Shu Zhang

**Affiliations:** 1Guangdong Provincial Key Laboratory of Advanced Drug Delivery Systems, Guangdong Pharmaceutical University, Guangzhou 510006, China; 2Biopharmaceutical R&D Center of Jinan University Co., Ltd., Guangzhou 510632, China

**Keywords:** transdermal drug delivery system, plant exosomes, nanoparticles, research progress

## Abstract

Exosomes act as emerging transdermal drug delivery vehicles with high deformability and excellent permeability, which can be used to deliver various small-molecule drugs and macromolecular drugs and increase the transdermal and dermal retention of drugs, improving the local efficacy and drug delivery compliance. At present, there are many studies on the use of plant exosome-like nanoparticles (PELNVs) as drug carriers. In this review, the source, extraction, isolation, and chemical composition of plant exosomes are reviewed, and the research progress on PELNVs as drug delivery systems in transdermal drug delivery systems in recent years has elucidated the broad application prospect of PELNVs.

## 1. Introduction

A transdermal drug delivery system (TDDS) is a drug delivery system in which drugs can penetrate the epidermis barrier, be absorbed by the skin, and then access the site of action. The drugs enter the skin at or near a constant rate, passing through the stratum corneum by a passive diffusion flow down a concentration gradient from the skin surface to the deeper layers of the skin. In this way, drugs reach the capillaries in the dermis, then enter the bloodstream, allowing them to circulate throughout the body or via local microcirculation to exert a certain therapeutic effect on the whole body (or locally) [1,2].

Nanoparticles are effective transporters in a TDDS. Due to their small particle sizes, they easily fuse with cells of the skin and promote drug penetration of the stratum corneum barrier without changing the skin’s structure. After, they establish drug storage on the skin’s surface and inside, presenting a local long-term therapeutic role [3]. Compared with traditional topical creams, nanoformulations can reduce the number of administrations and improve patient compliance; they have become a topic of interest in TDDSs.

Various nanoparticles, including liposomes, iron oxide nanoparticles, and polymeric nanoparticles, are being tested in clinical and preclinical studies for drug and gene delivery, with the goal of improving site-specific delivery [4]. Exosomes, as natural nanoparticles, have gained popularity in recent years. They are multivesicular bodies (MVBs) with diameters of approximately 30–150 nm, smaller than other MVBs, such as apoptotic vesicles, with diameters of approximately 800–5000 nm, and microvesicles at 200–1000 nm [5]. Exosomes are tea saucer-shaped or cup-shaped and have double-layered phospholipid membrane structures.

Exosomes are formed by budding into the lumen through the endosomal membrane. Multivesicular bodies fuse with plasma membranes and secrete out of cells, which are transmitted in vivo through biological fluids [6]. In the process of exosome transmission in vivo, they are internalized by receptor cells through receptor-mediated endocytosis, pinocytosis, phagocytosis, or direct release of the content into the cytoplasm by fusion with the cell membrane, thereby mediating communication with the recipient cells [6,7].

In the past few decades, extracellular vehicles (EVs) in plant cells (which have rigid walls) have attracted little attention compared to EVs in mammalian cells. However, in recent decades, increasing evidence indicates that plant cells also secrete EVs that play a key role in plant growth, development, and functioning, which have attracted researchers’ attention. Plant exosome-like nanoparticles (PELNVs) are biotechnology products with great potential for development. They have the advantages of easy extraction, high stability, good tissue penetration, and low immunogenicity [8], which make them have great clinical applications in DDSs and TDDSs. As potential excellent vehicles in TDDSs, PELNVs not only have active constituents in a parent-of-origin plan but can also be used to deliver exogenous substances, such as mRNA, miRNA, mitochondrial DNA, and proteins, which are far better than synthetic nanoparticles and mammal EVs. As we know, synthetic nanoparticles only serve the purpose of delivering drug molecules and do not contain any innate therapeutic capabilities.

Although mammal EVs also have innate therapeutic active ingredients, due to differences in species, immunological rejection is the key problem for their application. In addition, compared with mammal EVs, PELNVs do not harbor zoonotic or human pathogens. Thus, PELNVs also take the edge over mammalian cell-derived ELNVs with their more innocuous traits. Relying on their natural sources and compositions, PELNVs enjoy the perks of being undetected by the human immune system, thus achieving enhanced circulation periods and a higher bioavailability, which is also the best advantageous characteristic compared with artificially synthesized nanoparticles, such as liposomes; more effort should be made to improve their immune tolerance [9,10]. Studies show why PELNVs could protect cargo that is not to be degraded during circulation but may rely on the natural lipid bilayer membrane. The lipids in PELNVs are key protection factors of their functions. The natural lipid bilayer membrane means PELNVs could tolerate the effects of temperature, pH, simulated physiological environment, and sonication [11], which means that PELNVs have relative stability and exhibit beneficial processability characteristics. Chen et al. confirmed that the storage of fresh ginger slices or freshly isolated PELNVs at 80 ℃ had no significant effects on the IL-1β-inhibiting activities of ginger-derived ELNs [12]. The exosome-like nanoparticles derived from edible mulberry bark (MBELNs) conferred protection against colitis in a mouse model by promoting heat shock protein family A (Hsp70) member 8 (HSPA8)-mediated activation of the AhR signaling pathway [13]. Plant-derived exosomal microRNAs shape the gut microbiota [14]. All of them showed that PELNVs have a high tolerance to gastrointestinal enzymes and bile extracts, which give them good stability characteristics during the circulation periods and are particularly well-suited as drug delivery carriers.

The beneficial processability characteristics of PELNVs are expressed in two fundamental ways, one is that PELNVs themself can be modified by engineering, and the other is that lipids in PELNVs could be reassembled into newly-styled nanocarriers. Zhang et al. isolated PELNVs from ginger, extracted their lipids, and reassembled them into nanoparticles. The assembled nanoparticles were efficiently loaded with Dox (doxorubicin). At the same time, the assembled nanoparticles were modified with folic acid, which then targeted the delivery of Dox into the tumor and enhanced the inhibitory effect of Dox on tumor growth [15].

As research continues, PELNVs are slowly yielding their secrets. The natural sources and compositions of PELNVs make them stable and tolerant to enzymes, pH, and economics, with low immune tolerance, tending to industrial production, and making them widely used in TDDSs (Figure 1).

Moreover, although the original intended use of nanomedicine was to improve human health, nanoparticles can be purposely misused for other intentions, as many researchers have reported due to the soiling or maximizing of the toxicity of the nanoparticles. Recently, their potential toxicological effects on humans, animals, and the environment have received some attention [16]. In order to decrease the costs of nanodrug delivery vehicles, make them more effective in the body, promote a healthy environment, and reduce unintended use, new approaches and design principles are clearly needed for this field. As members of nanoparticles, natural PELNV-based nanoplatforms are innocuous, biocompatible, and able to be produced economically at a large scale. Natural PELNVs are the new hope for nanomedicine.

## 2. Origin, Extraction, and Isolation of PELNVs

### 2.1. Origin of Plants

PELNVs were first observed in the 1960s, with a TEM (transmission electron microscope) observation that the fusion of MVB with the plasma membrane led to the release of vesicles into the extracellular space of fungi and higher plants [17,18]. An et al. [19] were the first to investigate whether plant cells secreted exosomes and used TEM to obtain evidence of plant-derived exosomes. Recently, it was shown that nanoparticles can be isolated from plants, such as lemon [20], grape [21], ginger [12], tomato [22], sunflower, and rice, which are almost all edible plants or characteristic vegetable drugs. The PELNVs of edible fruit and vegetable origins enter the body at an early stage mainly through oral administration; they have protective effects against alcoholic liver injury and modulate intestinal flora [14].

### 2.2. Extraction and Isolation

At present, the methods commonly used for exosome separation mainly include ultracentrifugation [23], sucrose density gradient centrifugation [24], ultrafiltration [25], immunoaffinity [26], polymer precipitation [26], size exclusion chromatography [27], and microfluidic technology [28]. However, these commonly used separation methods have their advantages and disadvantages (Table 1).

Density gradient separation by differential ultracentrifugation has become the “gold standard” for separating PELNVs due to its low cost and ease of operation. Ultracentrifugation and sucrose density gradient centrifugation take advantage of the different sizes and densities of exosomes from other components to separate exosomes under a specific centrifugal force. This method is mature and inexpensive and can be used for large-volume samples; however, it is tedious and time-consuming to perform, recovery is unstable, and exosomes may be damaged. Sucrose density gradient centrifugation is mainly used in the purification of exosomes and can obtain high-purity exosomes. Ultrafiltration is a separation technique based on different particle sizes and molecular weights; it uses ultrafiltration membranes to separate exosomes from biological macromolecules, such as proteins. This method does not require special equipment and reagents, and it is simple to operate. However, in the process of ultrafiltration, it easily causes the loss of small particle-sized exosomes, the blockage of the filter membrane, and damage to exosomes caused by extrusion. The immunoaffinity method is based on the interaction between antibodies and the membrane protein on the surface of exosomes. This method has high specificity and purity for the separation of exosome subtypes; however, this method has a high cost and harsh operating conditions, which mainly depend on the specificity of antibodies, and the exosomes isolated by this method may lose their original characteristics. The polymer precipitation method uses highly hydrophilic polymers, such as polyethylene glycol to reduce the solubility or dispersion of exosomes. This method is simple and suitable for large-volume samples, but the presence of potential contaminants (coprecipitated protein aggregates or residual polymers) during the extraction process leads to a decrease in the exosome purity [29]. Size exclusion chromatography is also a separation technique used for particles of different sizes and molecular weights. This separation method is simple, economical, and has high separation purity, and it can maintain the biological function and structure of exosomes but requires special columns and fillers, and there is a risk of membrane protein contamination. In recent years, the widespread application of microfluidic technology has provided new ideas for the separation of exosomes. Based on the differences in the biochemical and physical properties of exosomes, microfluidics is highly sensitive and operates at a high speed for the separation of exosomes, but the yield is low, and it is only suitable for diagnosis. In addition, an increasing number of commercial kits have been developed for exosome isolation. This method is simple and convenient to operate without special equipment, and the extraction efficiency and purification of exosomes have gradually improved with the upgrading of products.

## 3. Physicochemical Characterization of PELNVs

Exosomes are carriers of bilayer membrane structures composed of proteins, lipids, and nucleic acids [30]. It is the existence of the vesicular membrane structure of exosomes that can protect their internal biomolecules from the influences of various enzymes in body fluids, thus maintaining their integrity and biological activity.

The chemical composition profiles of PELNVs maintain significant discrepancies from those of mammalian-derived exosomes in terms of their protein, lipid, and RNA content [31]. Compositional analyses of lipids, nucleic acids, and proteins of PELNVs are appreciated as important characterization criteria for the quality control of PELNVs [32,33]. Plant and mammalian cell-derived ELNVs share mutual common techniques used for chemical component characterization. (Table 2).

### 3.1. Identification of Proteins

The proteins that comprise exosomes mainly include the family of three transmembrane proteins (e.g., CD9, CD63, and CD81), antigen-presenting molecules (i.e., MHCI and MHCII), glycoproteins, adhesion molecules, heat shock proteins, cytoskeletal proteins, membrane transport, fusion proteins, ESCRT (endosomal-sorting complex required for transport), growth factors, cytokines, and some signaling receptors [34]. Exosomal proteins are mainly associated with exosome biogenesis, secretion, targeting, uptake, and signaling, and exosomal proteins of specific donor cell origin can be involved in antigen presentation to participate in the immune response [34,35,36]. Yuan Liu et al. optimized isolation methods for two types of plant vesicles: nanovesicles from disrupted leaves and sEVs from the extracellular apoplastic spaces of *Arabidopsis thaliana* and *Brassicaceae* vegetables. Both preparations yielded intact vesicles of uniform sizes and a mean membrane charge of approximately −25 mV. The proteomic analysis of a subset of vesicles with a density of 1.1–1.19 g/mL^−1^ sheds light on the likely cellular origin and complexity of the vesicles. *Kaempferia parviflora* extracellular vesicles (KPEVs) contained 5,7-dimethoxyflavone, the major bioactive compound of *Kaempferia parviflora* [57].

### 3.2. Identification of Lipids

Exosomes also contain a variety of lipids, such as cholesterol, ceramide, sphingomyelin, phosphatidyl alcohol (PI), phosphatidylserine (PS), phosphatidylcholine (PC), phosphatidylethanolamine (PE), and gangliosides (GMs). The presence of lipids mainly constitutes the rigid bilayer membranes of exosomes and affects cargo sorting as well as exosome secretion, structure, and signaling [41,42]. Stefania Raimondo et al. isolated PDEVs (plant-derived extracellular vesicles) from the juice of citrus limon L. (LEVs) and characterized their flavonoid, limonoid, and lipid contents [43]. They performed a metabolomic analysis on LEVs that identified 45 compounds. The detected compounds can be classified into different families, such as organic acids, flavonoids, limonoids, cinnamic acid derivatives, lysophospholipids, an acyl-thioester, carbohydrates, a phenolic acid derivative, and two nucleotide derivatives.

### 3.3. Identification of Nucleic Acids

In addition, exosomes contain nucleic acids, such as DNA, mRNA, miRNA, and noncoding RNA. Gel electrophoresis demonstrated the presence of substantial amounts of small-sized RNAs (fewer than 300 nucleotides) in GELN (ginger exosome-like nanoparticles) isolated from ginger tissue [14]. Next-generation sequencing analysis of the GELN RNA further suggested that GELN contained miRNA. At a sequencing depth of 20 million reads, 93,679 of the miRNA reads were mapped to 109 mature miRNAs from the NCBI plant miRNA database. The nucleic acid components of exosomes mainly play a role in various biological processes, such as exocytosis, hematopoiesis, and angiogenesis, and are involved in exosome-mediated cellular communication [47]. For example, miRNAs packaged in PDEVs can be the key modulators of human gene expression, representing potential therapeutic agents. Interestingly, concerning the phrase “an apple a day keeps the doctor away”, it would be mediated mainly by miRNA146 production by M2 macrophages when they are treated with exosomes derived from apples [48]. Now, miRNA-like small RNAs (milRNAs) have been explored [49]. An A. bisporus RNA-seq was studied in order to identify potential de novo milRNAs. Precursors and mature milRNAs were found in the edible parts (i.e., caps and stipes). When their potential gene targets were investigated, the results highlighted that most were involved in primary and secondary metabolic regulations. Next, the human transcriptome was used as a target, and the results suggest that those milRNAs might interfere with important biological processes related to cancer, infection, and neurodegenerative diseases.

Furthermore, small RNAs (sRNAs) were reported as a new class of functional components in PELNVs [58]. The sRNAs in PELNVs, from a decoction of Hong Jing Tian (HJT, RHODIOLAE CRENULATAE RADIX ET RHIZOMA, Rhodiola crenulate), could deliver therapeutic reagents in vivo. sRNAs, such as HJT-sRNA-m7 and PGY-sRNA-6 in the PELNVs, exhibited potent antifibrosis and anti-inflammatory effects, respectively.

### 3.4. Identification of Small Molecule Compound

Fresh plant materials, such as herbs, vegetables, fruits, and their extracts, are the cornucopia of many nutrients and contain a variety of small molecular compounds that are beneficial to human health. This is also the unique advantage of PELNVs. Wang et al. isolated exosomes from grapefruits containing a lot of naringin and naringenin, which have anti-inflammatory, anti-virus, anti-cancer, antimutagenic, anti-allergic, anti-ulcer, analgesic, and hypotensive activities [53]. Citric acid from lemon ELNVs could enhance the normal metabolism of the human body. Zhuang et al. [54] detected 6-gingerol from ginger ELNVs, which not only has good anti-tumor application prospects but also has a variety of effects, such as anti-inflammatory properties and inhibition of the central nervous system. Deng et al. [55] detected sulforaphane from broccoli exosomes, which have anti-tumor, detoxification, antibacterial, and antioxidant effects [56].

## 4. Study of PELNVs as Drug Delivery Systems

Drug delivery systems (DDSs) can deliver drugs from outside of the body to specific locations in the body via a carrier. It is a specific system capable of comprehensively regulating the spatial, temporal, and dose distribution of drugs in the body to increase the pharmacological activity of drugs and reduce their side effects [59,60,61]. The cellular and molecular accuracies of DDSs offer new possibilities for the diagnosis and treatment of diseases [62]. All DDSs are constructed to increase the therapeutic dose of the drug at the target site and to maintain a certain duration of action [60]. Nanoparticles, bacteria, and viruses are commonly used as delivery carriers to protect the drug from degradation and to target the delivery of the drug to the target site [63,64]. In recent years, the innovation of nanocarriers has promoted the development of DDSs; traditional nanodrug transport carriers can overcome the shortcomings of methotrexate (MTX), doxorubicin (DOX), paclitaxel (PTX), curcumin, etc., such as poor water solubility, easy rapid clearance by the body, and poor biocompatibility, and improve their drug solubility and dispersion. The penetration abilities of these nanocarriers to biological barriers are still weak, and the problems of stability, biocompatibility, and toxicity have not been effectively solved [65,66].

### 4.1. Methods of Loading Cargo by PELNVs

As natural nanoscale vesicle structures, PELNVs are potentially efficient carrier materials with good biocompatibility and safety, and they can cross biological barriers [67]. They can load long and short chains, coding, noncoding RNA (e.g., mRNA, miRNA, and lncRNA), lipids, proteins, and active chemotherapy drugs. The most frequently used methods to load cargo include chemical transfection, electroporation, ultrasound, co-incubation, extrusion, the freeze–thaw cycle, auxiliary saponins, low-osmosis dialysis, and pH gradients [68,69]. They rely on bilayer membrane structures to wrap water- and lipid-soluble drugs, reduce drug toxicity, and protect drugs [70].

### 4.2. Low Immunogenicity Makes PELNVs and Their Cargo More Medically Effective

The most unique advantage of PELNVs is that they cannot be detected by the immune system due to their natural origin and composition. Moreover, they do not carry zoonotic pathogens and have higher bioavailability. The gut mucosal immune system is considered to play an important role in counteracting the potential adverse effects of food-derived antigens, such as PELNVs. The nanovesicles isolated from grapefruits could be selectively taken up by intestinal macrophages, so they could colitis-target deliver MTX to intestinal cells and alleviate DSS (dextran sulfate sodium)-induced colitis [53]. The lemon nanovesicles suppress CML tumor growth in vivo by specifically reaching the tumor site; they then activate TRAIL-mediated apoptotic cell processes and inhibit the secretion of VEGF-A (vascular endothelial growth factor A), IL-6 (interleukin-6), and IL-8 (interleukin-8) to suppress angiogenesis [71].

### 4.3. Good Biocompatible of PELNVs

PELNVs show a fascinating biocompatible nature because of their natural provenance. We know that artificially synthesized nanoplatforms could induce an immune response in the host, meaning that they are sequestered by filtering through organs and, hence, clearance of these nanomaterials is enhanced as a result. There have been meager attempts to find cytotoxic effects of PELNVs on living subjects. For instance, to explore the potential cytotoxic effects of grapefruit-derived ELNVs in mice, several markers, such as pro-inflammatory cytokines and serum levels of liver enzymes, e.g., alanine transaminase (ALT) and aspartate transaminase (AST), were quantitatively measured. Mice were pre-administered with grapefruit-derived ELNVs and commercially available DOTAP (1,2-dioleoyl-3-trimethylammonium-propane)-DOPE (dioleoylphosphati-dylethanolamine) liposomes. ALT, AST, and pro-inflammatory cytokine levels were reported to be notably elevated in liposomal-treated mice; surprisingly, no elevation was recorded in the PELNV-treated mice group. In addition to this, no pathological modification or necrosis was observed in any of the histological samples of the liver, kidneys, spleen, and lungs in the PELNV-treated mice group [72]. It can be seen that PELNVs, as natural nanoscale drug delivery carriers, have broad application prospects.

## 5. Application of PELNVs in Transdermal Drug Delivery Systems

Plants have a long history of use in the treatment of skin cancer and skin disorders. PELNVs are secreted nanovesicle messengers responsible for intercellular communications and show promise as a new biotechnological skin care agent. PELNVs are naturally generated and carry innocuous components from their parent cells, some of which have been proven to be therapeutic. For instance, nanoparticle compositions containing EVs from ginseng, pine leaves, salvia, and other plant sources have been reported in the literature to show hair regrowth-promoting effects by deeply penetrating the skin and providing nutrients, stimulating hair follicles and exerting antioxidant activity on the scalp [73].

### 5.1. Skin Penetration Efficiency of PELNVs

PELNVs can intrinsically localize at target tissues—some of the most important traits of a targeted delivery system. It was reported that a broccoli-derived PELNV is highly lipophilic with high interception efficacy and can be used to penetrate deeply into the skin tissue, which was confirmed by the transport of encapsulated fluorescent agents in a broccoli-derived PELNV into keratinocytes and the presence of broccoli protein in the plasma membrane of keratinocytes [74]. Compared with the blank control group, the skin penetration efficiency of the cucumber-derived exosome vesicles was two times higher after mixing with lipophilic drug substitutes. They could significantly penetrate the dermis, providing theoretical support for the use of plant exosomal vesicles as a transdermal drug delivery system to deliver lipophilic drugs [75].

Furthermore, nucleic acid therapy, including therapy with sRNAs, has been considered the next wave of medicine. Although six sRNA drugs targeting disease-related genes have been approved by the FDA (Vitravene, Macugen, Kynamro, Exondys 51, Defitelio, and Spinraza), the drug delivery efficacy is not satisfactory. Considering the enhanced skin penetration and cargo sequestration efficacy, PELNVs can transdermally deliver genes for cutaneous melanoma based on past experiences in transdermal gene delivery [76].

### 5.2. Regenerative Medicine Applications

Various lesions of the skin, such as burns, sunburns, cuts, diabetic foot ulcers, and pressure ulcers, are all related to the dysfunction of the skin barrier, which is a public health problem. Regenerative medicine applications are believed to be the most effective approaches to repairing the skin barrier. Elahe Mahdipour [77] isolated exosomes from beet extracts and investigated the effects of beet exosomes on skin fibroblast migration and gene expression profiles. The results showed that beet exosomes have antioxidant and scavenging effects. The neovascularization potential of the vascular endothelial cells treated with beet exosomes increased. In addition, beet exosome treatment modulated the ability of fibroblasts to produce collagen I, collagen III, and hyaluronate synthase. Beet exosomes can also inhibit the migration capacity of fibroblasts. This finding provides a new cosmetic and therapeutic idea for beet exosomes. Fikrettin Şahin et al. [78] investigated the role of wheat exosomes in regenerative medicine applications or drug development and for the first time demonstrated the activity of wheat exosomes in the wound healing process using in vitro methods. Although the underlying wound healing process remains a mystery, in the current study, exosomes derived from wheatgrass juice were examined for the efficiency of cell survival and migration. When the concentration of wheat exosomes increased from 30 to 200 μg/mL, the proliferation and migration effects of wheat exosomes on the endothelial cells, epithelial cells, and dermal fibroblasts were surprising. In summary, this study proposes the proliferation and migration characteristics of wheat-derived exosomes, providing a new beginning for the treatment strategy of skin wound repair. Grapefruit-derived extracellular vesicles (GEVs) could be used as promising cell-free therapeutic tools for wound healing [79]. GEVs increased cell viability and cell migration while reducing the intracellular ROS (reactive oxygen species) production in HaCaT cells (human immortalized keratinocytes). The expressions of proliferation and migration-related genes were raised by GEV treatment in a dose-dependent manner. Additionally, GEV treatment increased the tube formation capabilities of treated HUVEC cells (human umbilical vein endothelial cells) (Figure 2).

## 6. Transdermal Mechanism of PELNVs

Valuable discoveries have been made concerning the mechanisms of action of PELNVs at the cellular and molecular levels. PELNV treatment promotes the proliferation and migration of fibroblasts, accelerates re-epithelialization by promoting the proliferation and migration of keratinocyte cells, and increases collagen synthesis and the production of elastin and fibronectin [80,81]. Nonetheless, the potential of PELNVs to improve the penetration efficacy of active pharmaceutical ingredients for derma applications has rarely been demonstrated; more detailed investigations are needed in this regard.

### 6.1. Cross-Kingdom Communications Improve the Penetration Efficacy

The primary role of PELNVs is to convey information to the recipient cells, affecting their functions [82]. This means that PELNVs would be the key to cross-kingdom communications between plants and mammalians, which may be one of the reasons why PELNVs improve the penetration efficacy of active pharmaceutical ingredients. In particular, the presence of microRNAs (miRNAs) in PDEVs represents an interesting aspect for understanding how PELNVs can target the mammalian genes involved in pathological conditions, such as skin diseases, cancer, inflammation, and oxidative stress.

Furthermore, PELNVs are structurally similar to liposomes and have high similarity to mammalian cell membrane surfaces; that is, all of them have bilayer phospholipid structures [83]. Thus, PELNVs can effectively cross the skin stratum corneum via trans- and intercellular pathways through lipid fusion effects. The pathways involved in the cellular internalization of PELNVs include phagocytosis, macropinocytosis, and clathrin-mediated endocytosis [53]. These findings suggest that the metabolic fate of PELNVs was possibly internalized via the phagocytosis pathway (Figure 3).

### 6.2. Penetration through the Trans-Adnexal Route

When applied topically to the skin, PELNVs could penetrate a fat-rich channel on the hair follicle, allowing for drug delivery [84,85]. Apart from the trans-epidermal route, there is another route, the trans-adnexal route, in which the drug is absorbed through hair follicles, sebaceous glands, and sweat glands [86]. Studies have found that PELNVs can reach the hair shaft in two ways [87]. In the first way, PELNVs enter the hair stromal cells and are transferred deep into the skin by cellular differentiation. Another way is that PELNVs enter the other side of the hair shaft directly from the tip or one side of the hair shaft (Figure 4) [88]. Many current studies have confirmed that PELNVs can penetrate the skin through the accessory apparatus pathway across the skin [89,90,91].

## 7. Conclusions and Perspectives

According to previous reports, ultracentrifugation-based exosome isolation is one of the most commonly used and reported techniques in PELNV isolation. It is estimated that ultracentrifugation accounts for 56% of all exosome isolation techniques employed by users in exosome research [92]. This approach is often perceived by many as easy to use, requiring very little technical expertise; it is affordable over time (i.e., requiring one ultracentrifuge machine for long-term use) and moderately time-consuming with little or no sample pretreatments. For these reasons, ultracentrifugation-based techniques have become very prevalent options among researchers in exosome research. There are two types of preparative ultracentrifugation: differential ultracentrifugation and density gradient ultracentrifugation.

Compared with other transdermal agents, PELNVs have the following advantages: low costs, high stability, good tissue penetration, and low immunogenicity. They also have unique advantages in the field of transdermal drug delivery, representing prospective biotechnology products with great developmental potential and broad prospects. In particular, PELNVs can be used as effective carriers for transdermal drugs, providing effective schemes for the transdermal delivery of such drugs.

At present, as an excellent drug carrier, PELNVs have been well studied in oral drug delivery and drug-targeted delivery, but PELNVs as drug carriers have been studied less in transdermal drug delivery systems. It can be foreseen that a PELNV delivery system with high stability, high transdermal volume, and ease of use will be developed for this excellent carrier in the future.

## Figures and Tables

**Figure 1 bioengineering-10-00104-f001:**
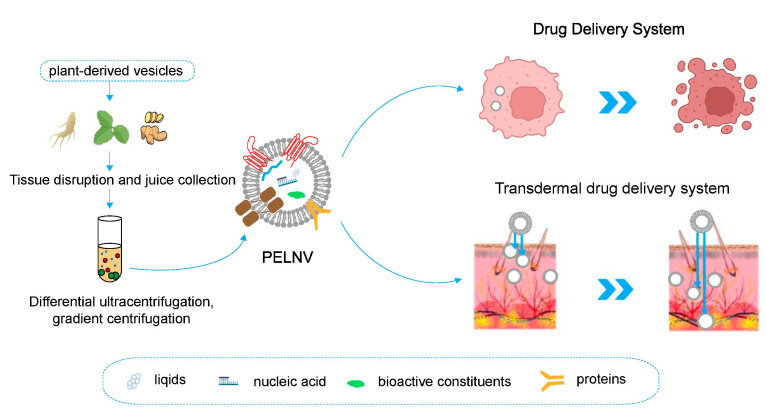
Overview of biological functions of PELNVs and their therapeutic applications.

**Figure 2 bioengineering-10-00104-f002:**
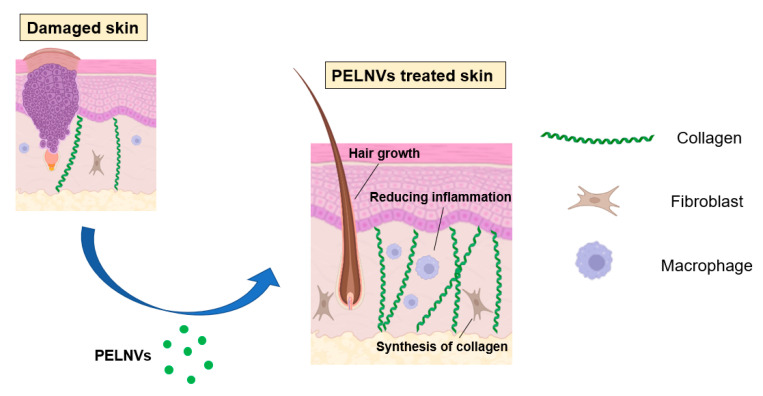
PELNVs used to repair the skin barrier.

**Figure 3 bioengineering-10-00104-f003:**
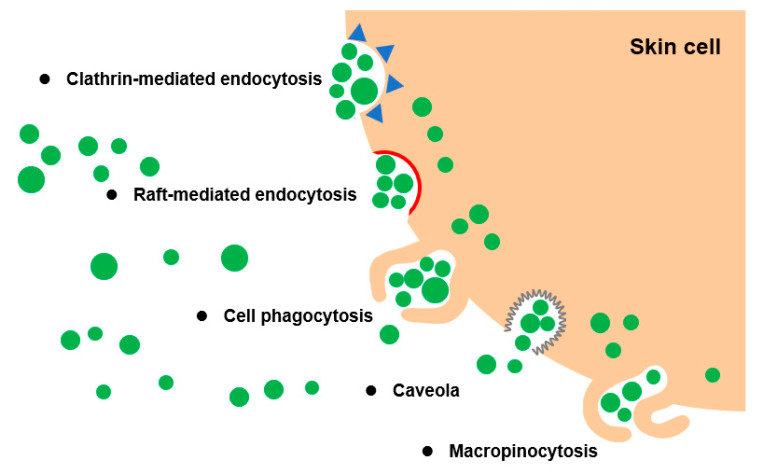
Cellular internalization of PELNVs by phagocytosis pathway.

**Figure 4 bioengineering-10-00104-f004:**
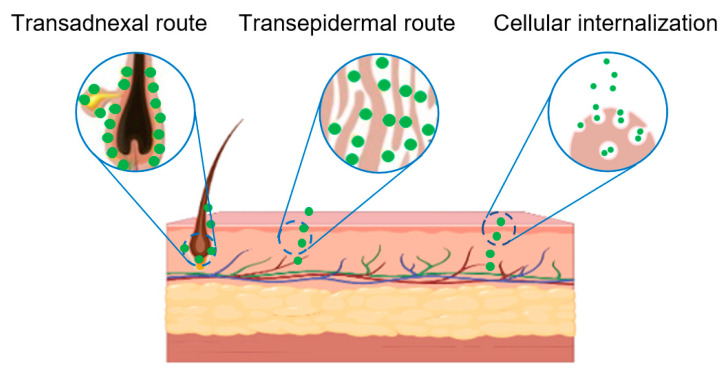
Schematic diagram of the transdermal administration of PELNV.

**Table 1 bioengineering-10-00104-t001:** Comparison of the advantages and disadvantages of common exosome extraction methods.

Method	Advantages	Disadvantages	Reference
Ultracentrifugation	Low cost, simple operation, the “gold standard” for separation, can be used for large-volume samples	Requires expensive instrumentation, time-consuming super ionization, unstable recovery rate, and possible damage to exosomes	[23]
Sucrose density gradient centrifugation	Low cost, not time-consuming, high purity of extracted exosomes, applicable to large-volume samples	Easily damages exosomes, requires certain manipulation	[24]
Ultrafiltration	Easy to operate, no need for expensive instruments	Time-consuming, easy to clog and cause loss, easily damages exosomes	[25]
Immunoaffinity method	High specificity, high purity of extracted exosomes	High cost and harsh conditions of use, not suitable for large-volume samples	[26]
Polymer precipitation method	Simple operation, no need for expensive instruments, stable yield, suitable for large-volume samples	Higher cost, slightly poorer purity of extracted exosomes, easily causes exosome aggregation	[26]
Size exclusion chromatography	Simple, economical, and high separation purity	Requires special fillers, time-consuming, and risk of contaminated protein	[27]
Microfluidics	High sensitivity, high speed	Poor applicability, currently only used for diagnosis, small processing capacity	[28]

**Table 2 bioengineering-10-00104-t002:** Composition of exosomes in plants and animals.

		Composition	Function	References
**Proteins**	Plants	Transmembrane proteins, antigen-presenting molecules, glycoproteins, adhesion molecules, heat shock proteins, cytoskeletal proteins, membrane transport, fusion proteins, ESCRT, growth factors, cytokines, and some signaling receptors	Exosome biogenesis, secretion, targeting, uptake, and signaling	[34,35,36]
	Animals	Membrane transport protein, fusion protein, heat shock protein, tetraspanins, multivesicular body biosynthesis-associated protein, cytoskeletal protein, signal transduction protein, and carrier protein	Transmembrane transport, biosynthesis, metabolism	[37,38,39,40]
**Lipid** **s**	Plants	Cholesterol, ceramide, sphingomyelin, phosphatidyl alcohol (PI), phosphatidylserine (PS), phosphatidylcholine (PC), phosphatidylethanolamine (PE), and gangliosides (GMs)	Constitute the rigid bilayer membrane of exosomes and affects cargo sorting as well as exosome secretion, structure, and signaling	[41,42,43]
	Animals	Cholesterol, diglycerides, sphingomyelin and ceramides, phospholipids, phosphatidylcholine (PC), phosphatidylserine (PS), phosphatidylethanolamine (PE), phosphatidylinositol (PI), and polyglycerol	Play an important role in the biological function of animal exosomes	[44,45,46]
**Nucleic** **ACIDS**	Plants	DNA, mRNA, miRNA, and non-coding RNA	Play a role in various biological processes, such as exocytosis, hematopoiesis, and angiogenesis, and are involved in exosome-mediated cellular communication	[14,47,48,49]
	Animals	DNA, mRNA, miRNA, mitochondrial DNA (mtDNA), piRNA, long noncoding RNA (lncRNA), rRNA, snRNA, snoRNA, and tRNA	Linked to some related diseases and can be used for cancer diagnosis	[39,50,51,52]
**Small molecule compound**	Plants	Naringin (grapefruit), naringenin (grapefruit), citric acid (lemon), sulforaphane (broccoli), 6-shogaol (ginger), etc.	Has a potential therapeutic effect	[53,54,55,56]

## Data Availability

Data sharing not applicable.

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
