# Peer review of "Plant Exosome-like Nanoparticles as Biological Shuttles for Transdermal Drug Delivery"

_bioengineering, 2023, doi:10.3390/bioengineering10010104_

Round 1
Reviewer 1 Report
The manuscript is well structured. More than 85% of the bibliographic references are adequate and up-to-date. But, some changes are needed to reach its optimal form.
In general, I noticed that the authors tend to use very long sentences. I recommend replacing them with shorter phrases, easy to read and understand. For example, in Introduction, Line 21: the first sentence "The transdermal drug delivery system (TDDS) is a drug delivery system in which..." is too long, it should be divided into two smaller sentences, because it is difficult to understand, the idea is lost until the end.
- Lines 34-38: After the presentation of the structure of exosomes, I recommend the introduction of a picture that illustrates exactly their structure, considering that the exosomes are the subject of the current review, but also so that those who read have a more accurate vision of them.
- Pay close attention to the abbreviations used for the first time in the text, it must be mentioned in brackets what they represent (TEM, Dox, ESCRT, PDEVs, GELN, NCBI...)
- Line 64: The phrase "The nanovesicles are tea saucer-shaped or cup-shaped and have a phospholipid bilayer structure similar to that of mammalian-derived exosomes [14]" describes the shape of these vesicles, so it must be removed from “Sources of plant exosomes” and put it in “Introduction” when the authors describe the exosomes structure (for example, after Line 38).
Table 1: "Advantage" and "Dezadvantage" should be put in the plural form, because there are several advantages and disadvantages, respectively, for each method.
Line 148: attention to abbreviations: GELNs or GELN?
Line 149: miRNA or miRNAs?
Line 157: Correct is “miRNAs”, and not “milRNAs”;
Line 164: Table 2 should be simplified, considering that the authors have described in detail, inside the text, the composition of exosomes. Everything written in the Table 2 is also found in the text! What is the purpose of the table then? So it has to be simplified, by pointing out only the essential things.
Line 202: it should be mentioned in the parenthesis what these listed terms represent - VEGF-A, IL-6 and IL-8?
Lines 203-207: the phrase “Our laboratory used classical differential centrifugation combined with… carriers carrying chemotherapeutic drugs [52]” is too long, must be splitted.
Line 207: A semicolon is used before the phrase "nanovesicles isolated from grapefruit could..." Why? It is a sentence that is not related to the authors' own research presented before it, so it should be written as an independent sentence.
Line 208: what does "DSS" mean?
Linia 210: once again, be careful with abbreviations! PELNVs or PELNs?
Lines 226-231: the phrase “Abraham M. Abraham et al. [56] used the classical separation and purification…” is too long, very difficult to follow; it must be splitted into 2-3 shorter sentences!
Line 244: “When the concentration of wheat exosomes was increased to 200 μg/mL…” - increased from zero or from a certain amount? Some details should be given here.
Lines 249-254: GEVs sau GEV? Ce inseamna ROS, HaCaT, HU-VEC?
Line 270-275: the paragraph “Elahe Mahdipour isolated exosomes from Beta vulgaris extract … PELNVs - BEX treatment inhibited the migration abilities of fibroblasts” trebuie sters pentru ca se repeta si mai sus intre 232-238.
Line 295: Conclusions: The conclusions should be further developed. I recommend adding a conclusion regarding the best method of isolating the exosomes from plants, with an emphasis on its efficiency and cost. And also a conclusion regarding the applications of the exosomes in drug delivery, pointing out the advantage of these transport vehicles compared to the others.
Author Response
Response to Reviewer 1 Comments
Point 1: In general, I noticed that the authors tend to use very long sentences. I recommend replacing them with shorter phrases, easy to read and understand. For example, in Introduction, Line 21: the first sentence "The transdermal drug delivery system (TDDS) is a drug delivery system in which..." is too long, it should be divided into two smaller sentences, because it is difficult to understand, the idea is lost until the end.
Response 1: Thanks to the reviewer’s advice. According to your suggestion, we have divided this sentence into three shorter parts. “A transdermal drug delivery system (TDDS) is a drug delivery system in which drugs can penetrate the epidermis barrier, be absorbed by the skin, and then access their site of action. The drugs enter the skin at or near a constant rate, passing through the stratum corneum by a passive diffusion flow down a concentration gradient from the skin surface to the deeper layers of the skin. This way, drugs reach the capillaries in the dermis, then enter the bloodstream allowing them to circulate throughout the body or local microcirculation to exert a certain therapeutic effect on the whole body or locally [1,2].”
Point 2: - Lines 34-38: After the presentation of the structure of exosomes, I recommend the introduction of a picture that illustrates exactly their structure, considering that the exosomes are the subject of the current review, but also so that those who read have a more accurate vision of them.
Response 2: Thanks to the reviewer’s advice. According to your suggestion, we have inserted a picture to illustrate the theme of this review.
Point 3: - Pay close attention to the abbreviations used for the first time in the text, it must be mentioned in brackets what they represent (TEM, Dox, ESCRT, PDEVs, GELN, NCBI...)
Response 3: Thanks to the reviewer’s advice. According to your suggestion, we have given the full spelling of the abbreviations at the first time appeared.
Point 4: - Line 64: The phrase "The nanovesicles are tea saucer-shaped or cup-shaped and have a phospholipid bilayer structure similar to that of mammalian-derived exosomes [14]" describes the shape of these vesicles, so it must be removed from “Sources of plant exosomes” and put it in “Introduction” when the authors describe the exosomes structure (for example, after Line 38).
Response 4: Thanks to the reviewer’s advice. According to your suggestion, we removed “the describes of the shape of these vesicles” from the “Sources of plant exosomes” and put it in “Introduction” .
Point 5: Table 1: "Advantage" and "Dezadvantage" should be put in the plural form, because there are several advantages and disadvantages, respectively, for each method.
Response 5: Thanks to the reviewer’s advice. According to your suggestion, we have corrected this part of the original text.
Point 6: Line 148: attention to abbreviations: GELNs or GELN?
Response 6: Thanks to the reviewer’s question. “GELN” is the correct writing. We have unified the wording of singular and plural throughout the text.
Point 7: Line 149: miRNA or miRNAs?
Response 7: Thanks to the reviewer’s question. We have unified the word of miRNA, and plural throughout the text.
Point 8: Line 157: Correct is “miRNAs”, and not “milRNAs”;
Response 8: Thanks to the reviewer’s question. “milRNAs” is an abbreviation for miRNA-like small RNAs, rather than microRNAs (mRNAs).
Point 9: Line 164: Table 2 should be simplified, considering that the authors have described in detail, inside the text, the composition of exosomes. Everything written in the Table 2 is also found in the text! What is the purpose of the table then? So it has to be simplified, by pointing out only the essential things.
Response 9: Thanks to the reviewer’s question. According to your suggestion, we have simplified Table 2.
Table 2. Composition of exosomes in plants and animals.
|
|
|
Composition |
Function |
References |
|
Proteins |
Plants |
Transmembrane proteins, antigen-presenting molecules, glycoproteins, adhesion molecules, heat shock proteins, cytoskeletal proteins, membrane transport and fusion proteins, ESCRT, growth factors, cytokines, and some signaling receptors |
Exosome biogenesis, secretion, targeting, uptake, and signaling |
[31-33] |
|
Animals |
Membrane transport protein, fusion protein, heat shock protein, tetraspanins, and multivesicular body biosynthesis-associated protein, cytoskeletal protein, signal transduction protein, and carrier protein |
Transmembrane transport, biosynthesis, metabolism |
[41-44] |
|
|
Lipids |
Plants |
Cholesterol, ceramide, sphingomyelin, phosphatidyl alcohol (PI), phosphatidylserine (PS), phosphatidylcholine (PC), phosphatidylethanolamine (PE), and gangliosides (GMs) |
Constitute the rigid bilayer membrane of exosomes and affects cargo sorting as well as exosome secretion, structure, and signaling |
[35-37] |
|
Animals |
Cholesterol, diglycerides, sphingomyelin and ceramides, phospholipids, phosphatidylcholine (PC), phosphatidylserine (PS), phosphatidylethanolamine (PE), phosphatidylinositol (PI), and polyglycerol
|
Play an important role in the biological function of animal exosomes |
[45-47] |
|
|
Nucleic acids |
Plants |
DNA, mRNA, miRNA, and non-coding RNA |
Play a role in various biological processes, such as exocytosis, hematopoiesis, and angiogenesis, and are involved in exosome-mediated cellular communication |
[22,38-40] |
|
Animals |
DNA, mRNA, miRNA, mitochondrial DNA (mtDNA), piRNA, long noncoding RNA (lncRNA), rRNA, snRNA, snoRNA, and tRNA |
Linked to some related diseases and can be used for cancer diagnosis |
[43,48-50] |
|
|
Small molecule compound |
Plants |
Naringin (grapefruit), naringenin (grapefruit), aitric acid (lemon), sulforaphane (broccoli), 6-shogaol (ginger), etc. |
Has a potential therapeutic effect |
[51-54] |
Point 10: Line 202: it should be mentioned in the parenthesis what these listed terms represent - VEGF-A, IL-6 and IL-8?
Response 10: Thanks to the reviewer’s advice. According to your suggestion, we have given the full spelling of the abbreviations at the first time they occur.
Point 11: Lines 203-207: the phrase “Our laboratory used classical differential centrifugation combined with… carriers carrying chemotherapeutic drugs [52]” is too long, must be splitted.
Response 11: Thanks to the reviewer’s question. According to your suggestion, we have divided this sentence into two shorter parts. “Our laboratory used classical differential centrifugation combined with ultracentrifugation to extract PELNVs from turmeric rhizome juice and used them as a delivery system for the mansonone derivative b16. We prepared the PELNVs-loaded b16 by the co-incubation method to achieve the construction of a delivery system for natural edible plant-derived carriers carrying chemotherapeutic drugs [70].”
Point 12: Line 207: A semicolon is used before the phrase "nanovesicles isolated from grapefruit could..." Why? It is a sentence that is not related to the authors' own research presented before it, so it should be written as an independent sentence.
Response 12: Thanks to the reviewer’s question. We have changed semicolon to period.
Point 13: Line 208: what does "DSS" mean?
Response 13: Thanks to the reviewer’s advice. DSS is the abbreviation of dextran sulfate sodium.
Point 14: Linia 210: once again, be careful with abbreviations! PELNVs or PELNs?
Response 14: Thanks to the reviewer’s question. “PELNVs” is the correct writing. We have unified the wording of singular and plural throughout the text.
Point 15: Lines 226-231: the phrase “Abraham M. Abraham et al. [56] used the classical separation and purification…” is too long, very difficult to follow; it must be splitted into 2-3 shorter sentences!
Response 15: Thanks to the reviewer’s advice. According to your suggestion, we have divided this sentence into two shorter parts. “Abraham M. Abraham et al. [73] used the classical separation and purification method to obtain cucumber-derived EVs, and compared with the blank control group, the skin penetration efficiency of the cucumber-derived exosome vesicles was two times higher after mixing with lipophilic drug substitutes. They could significantly penetrate into the dermis, providing theoretical support for the use of plant exosomal vesicles as a transdermal drug delivery system to deliver lipophilic drugs.”
Point 16: Line 244: “When the concentration of wheat exosomes was increased to 200 μg/mL…” - increased from zero or from a certain amount? Some details should be given here.
Response 16: Thanks to the reviewer’s question. According to your suggestion, we have recognized our mistakes and have searched the corresponding literature and made supplements. “When the concentration of wheat exosomes increased from 30 to 200 μg/mL, the proliferation and migration effects of wheat exosomes on the endothelial cells, epithelial cells, and dermal fibroblasts were surprising.”
Point 17: Lines 249-254: GEVs sau GEV? Ce inseamna ROS, HaCaT, HU-VEC?
Response 17: Thanks to the reviewer’s question. “GEVs” is the correct writing. According to your suggestion, we have given the full spelling of the abbreviations at the first time they occur.
Point 18: Line 270-275: the paragraph “Elahe Mahdipour isolated exosomes from Beta vulgaris extract … PELNVs - BEX treatment inhibited the migration abilities of fibroblasts” trebuie sters pentru ca se repeta si mai sus intre 232-238.
Response 18: Thanks to the reviewer’s question. According to your suggestion, we have removed this part.
Point 19: Line 295: Conclusions: The conclusions should be further developed. I recommend adding a conclusion regarding the best method of isolating the exosomes from plants, with an emphasis on its efficiency and cost. And also a conclusion regarding the applications of the exosomes in drug delivery, pointing out the advantage of these transport vehicles compared to the others.
Response 19: Thanks to the reviewer’s question. According to your suggestion, we have greatly enriched the manuscript.

Reviewer 2 Report
The current review article discussed the use of plant exosome-like nanoparticles for the transdermal drug delivery. The manuscript is well-written and the writing is easy to follow. However, few comments need to be addressed before its publication.
1. More details should be added to the introduction. The introduction should be more informative and supported with more references.
2. Slight English editing should be carried out.
3. A figure illustrating the main idea of the review article will be highly appreciated.
Author Response
Response to Reviewer 2 Comments
Point 1: More details should be added to the introduction. The introduction should be more informative and supported with more references.
Response 1: Thanks to the reviewer’s question. According to your suggestion, we have added related content.
Point 2: Slight English editing should be carried out.
Response 2: Thanks to the reviewer’s question. According to your suggestion, we have made language edits. For your convenience, we have attached the English editing certificate.
Point 3: A figure illustrating the main idea of the review article will be highly appreciated.
Response 3: Thanks to the reviewer’s question. According to your suggestion, we have added a graphical abstract.

Reviewer 3 Report
The review is somehow confusing; authors need to very well distinguish between the animal sourced and plants sourced vesicles. It does not provide clear cut differentiation, first leads as an animal product and later goes for the plants and at times mixed idea and narrations.
Also, the review is not comprehensive in nature, no full information on all relevant sub-topics is available, and few figures are there, all sections including section 7 needs representative figures. Introduction is very short, and there is no discussion of skin delivery concepts, requirements, comparison with other nano-scale vesicles, pharmacokinetics details, delivery profile, payload distribution patterns, biocompatibility and biodegradability studies, metabolic fate, etc.
Authors need to provide a table of plants- based exosomes examples containing info on method of action. mechanism, delivery kinetics, disease/conditions of use and references.
The review seems to be prepared in haste! Needs major changes and populated with information.
Author Response
Response to Reviewer 3 Comments
Point 1: The review is somehow confusing; authors need to very well distinguish between the animal sourced and plants sourced vesicles. It does not provide clear cut differentiation, first leads as an animal product and later goes for the plants and at times mixed idea and narrations.
Response 1: Thanks to the reviewer’s question. We have tried to separate animal and plant exosomes related content. At present, most of the studies are on animal exosomes, and there are few studies on plant exosomes and our review aims to provide a basic reference for future research. So we quoted some content exosomes from animals as a supplement.
Point 2: Also, the review is not comprehensive in nature, no full information on all relevant sub-topics is available, and few figures are there, all sections including section 7 needs representative figures. Introduction is very short, and there is no discussion of skin delivery concepts, requirements, comparison with other nano-scale vesicles, pharmacokinetics details, delivery profile, payload distribution patterns, biocompatibility and biodegradability studies, metabolic fate, etc.
Response 2: Thanks to the reviewer’s question. According to your suggestion, we have augmented the introduction section and added images to illustrate our overview.
Point 3: Authors need to provide a table of plants-based exosomes examples containing info on method of action. mechanism, delivery kinetics, disease/conditions of use and references.
Response 3: Thanks to the reviewer’s question. The research on pharmacokinetics is more focused on intravenous injection and other administration methods, while the research on transdermal absorption is less. What a pity that no references had been be retrieved by us.

Round 2
Reviewer 3 Report
Manuscript has been modified, but it still lacks the depth which was recommended. Instead of explaining their points as replies to reviewer and editor, the authors need to incorporate the information or lack of information in the ms at the appropriate level under exact and suitable heading/sub_heading. Make sure all the comments are followed fully.
Author Response
Point:
Manuscript has been modified, but it still lacks the depth which was recommended. Instead of explaining their points as replies to reviewer and editor, the authors need to incorporate the information or lack of information in the ms at the appropriate level under exact and suitable heading/sub_heading. Make sure all the comments are followed fully.
Response:
Thanks to the reviewer’s question. Many thanks for giving me the second revision opportunity. I'm sorry because I didn't revise carefully on round 1 because I contracted COVID-19 unfortunately. Hope it does not cause you any inconvenience.
We has considered your comments and suggestions carefully once again, and revised the text. I have done my best to improve it and wish you satisfied with it. We've improved the introduction and emphasized the advantage of the PELNVs. Compared with synthetic nanovesicles and mammal exosomes, they have a lot of advantages. Such as convenient extraction, low immunogenicity, high stability, etc. Moreover, based on the original manuscript, we have further added we have supplemented the mechanism, application and study of PELNVs as drug delivery systems. Due to too much modified contents, I apologize for the inconvenience but would you mind to read the revised manuscript?
In addition, we have made further revisions on round 1 of your comments. For your convenience, we have attached the supplemental content on the next page.
Point 1:
The review is somehow confusing; authors need to very well distinguish between the animal sourced and plants sourced vesicles. It does not provide clear cut differentiation, first leads as an animal product and later goes for the plants and at times mixed idea and narrations.
Response 1:
Thanks to the reviewer’s advice. In the full text, especially introduction and the part 2, we further differentiated the PELNVs and the animal sourced exosomes to emphasizing the benefits of PELNVs, including but are not limited to convenient extraction, low immunogenicity, high stability, etc. Furthermore, we further added the comparison of origin and physicochemical characterization of animal sourced exosomes and PELNVs to better illustrate their differences. Please see Table 2.
Point 2:
Also, the review is not comprehensive in nature, no full information on all relevant sub-topics is available, and few figures are there, all sections including section 7 needs representative figures. Introduction is very short, and there is no discussion of skin delivery concepts, requirements, comparison with other nano-scale vesicles, pharmacokinetics details, delivery profile, payload distribution patterns, biocompatibility and biodegradability studies, metabolic fate, etc.
Response 2:
Thanks to the reviewer’s question.
Firstly, according to your suggestion, we have included relevant sub-topics where appropriate to better illustrate our content. For your convenience, we have attached a comparative table on the next page.
Comparative table
|
Previous heading |
Current heading |
|
1. Introduction |
1.Introduction |
|
2. Sources of PELNVs |
2. Origin, extraction and isolation of PELNVs |
|
3. Extraction and isolation of PELNVs |
2.1 Origin of plant |
|
|
2.2 Extraction and isolation |
|
4. Physicochemical characterization of PELNVs |
3. Physicochemical characterization of PELNVs |
|
|
3.1 Identification of proteins |
|
|
3.2 Identification of lipids |
|
|
3.3 Identification of nucleic acids |
|
|
3.4 Identification of small molecule compound |
|
5. Study of PELNVs as drug delivery systems |
4. Study of PELNVs as drug delivery systems |
|
|
4.1 Methods of loading cargo by PELNVs |
|
|
4.2 Low immunogenicity makes PELNVs and its cargo more medical effectiveness |
|
|
4.3 Good Biocompatible of PELNVs |
|
6. Application of PELNVs in transdermal drug delivery systems |
5. Application of PELNVs in transdermal drug delivery systems |
|
|
5.1 Skin penetration efficiency of PELNVs |
|
|
5.2 Regenerative medicine applications |
|
7. Transdermal mechanism of PELNVs |
6. Transdermal mechanism of PELNVs |
|
|
6.1 Cross-kingdom communication improve the penetration efficacy |
|
|
6.2 Penetration through trans adnexal route |
|
8. Conclusions and Perspectives |
7. Conclusions and Perspectives |
Secondly, we have added figures to illustrate our review. For your convenience, we have attached the figures.
Finally, we have supplied in the lack information: We have added the content of low immunogenicity and good biocompatible of PELNVs in section 4; the content of skin penetration efficiency and regenerative medicine applications in section 5; the content of penetration efficacy and metabolic fate in section 6. Due to too much modified contents, I apologize for the inconvenience but would you mind to read the revised manuscript?
Point 3:
Authors need to provide a table of plants- based exosomes examples containing info on method of action. mechanism, delivery kinetics, disease/conditions of use and references.
Response 3:
Thanks to the reviewer’s question, but we require to discuss with you. After a lot of searching, we found that there is not enough research on the transdermal administration of PELNVs to make a whole table. Especially there are few research of pharmacokinetics on transdermal absorption. We hope future scientist can do more research on this in the future.
In addition, we have made language edits. For your convenience, we have attached the English editing certificate.
